

# Assessing the accuracy of remotely-sensed fire datasets across the Southwestern Mediterranean basin

Luiz Felipe Galizia[1], Thomas Curt[1], Renaud Barbero[1], Marcos Rodrigues[2,3]

[1]INRAE, Mediterranean Ecosystems and Risks, Aix-en-Provence, France.
[2]Department of Agricultural and Forest Engineering, University of Lleida, Lleida, Spain.
[3]Joint Research Unit CTFC-AGROTECNIO, Solsona, Lleida, Spain

*Correspondence to*: Luiz Felipe Galizia (luiz.galizia@inrae.fr)

**Abstract.** Recently, many remote-sensing (RS) based datasets providing features of individual fire events from gridded global burned area products have been released. Although very promising, these datasets still lack a quantitative estimate of their accuracy with respect to historical ground-based fire databases. Here, we compared three state-of-the-art RS datasets (Fire Atlas, FRY and GlobFire) with high-quality ground databases compiled by regional fire agencies (AG) across the Southwestern Mediterranean basin (2005-2015). We assessed the spatial and temporal accuracy in estimated RS burned area (BA) and number of fires (NF) aggregated at monthly and 0.25° resolutions, considering different individual fire size thresholds ranging from 1 to 500 ha. Our results show that RS datasets were highly correlated with AG in terms of monthly BA and NF but severely underestimated both (by 38% and 96%, respectively) when considering all fires > 1 ha. Stronger agreement was found when increasing the fire size threshold, with fires > 100 ha denoting higher correlation and much lower error (BA 10%; NF 35%). The agreement between RS and AG was also the highest during the warm season (May to October) in particular across the regions with greater fire activity such as the Northern Iberian Peninsula. The Fire Atlas displayed a slightly better performance, with a lower relative error, although uncertainty in gridded BA product largely outpaced uncertainties across the RS datasets. Overall, our findings suggest a reasonable agreement between RS and ground-based datasets for fires larger than 100 ha, but care is needed when examining smaller fires at regional scales.

## 1 Introduction

Vegetation fires are a common and destructive hazard in the Southwestern Mediterranean basin. Over the last four decades, about 47,766 fires were responsible every year for 413,209 ha burned in this region (San-Miguel-Ayanz et al., 2017) causing extensive economic and ecological losses, and even human casualties (Keeley et al., 2011; Molina-Terrén et al., 2019). Fire is a complex phenomenon due to the confluence of several factors including climate, weather, human activities and vegetation (Bowman et al., 2009). The Mediterranean fire regime is dominated by human-caused ignitions (Ganteaume et al., 2013) with most of the total burned area (BA) linked to a small number of large fires during the summer (Turco et al., 2016). These large fire events are facilitated by dry conditions and high temperatures, which are both expected to increase in the future under the





ongoing climate change (Dupuy et al., 2020; Ruffault et al., 2020; Turco et al., 2018a). Additional factors such as landscape changes as well as changes in forest and fire management may also shape future fire activity (Moreira et al., 2020; Pausas and Fernández-Muñoz, 2012). However, there is still much uncertainty in the projected change of fire activity, and modeling efforts

across broad geographical scales are needed to better understand processes and mechanisms conductive to fire ignition and spread.

One of the main limitations in fire modeling lies in the lack of reliable and harmonized information on fire activity (Hantson et al., 2016; Williams and Abatzoglou, 2016). This is particularly true in Europe where the lack of data sharing as well as the lack of consistent quality-control procedures of national ground-based fire datasets has hampered analysis of fire regimes

across broader regional or continental scales (Mouillot and Field, 2005; Turco et al., 2016). To overcome this challenge, the European Forest Fire Information System (EFFIS; San-Miguel-Ayanz et al., 2015) is increasingly relying on remote-sensing (RS) techniques for monitoring fire incidence across Europe.

In the last decade, RS has contributed to 'fill the gap of knowledge' fostering fire-related products with spatial and temporal consistency, and global coverage (Chuvieco et al., 2019; Mouillot et al., 2014). The MODIS sensor outstands as one of the

best data provider for most applications such as MCD64A1 (Giglio et al., 2018) and FireCCI50 (Chuvieco et al., 2018). In particular, the latest generation of BA mapping products, the MCD64A1v006, sets the basis for an exhaustive global estimation of fire-related carbon emissions, compiled in the GFED4 database (Giglio et al., 2013; Randerson et al., 2015; van der Werf et al., 2017). Although BA products typically offer information about the pixels that burned in a given day, they do not provide information such as starting/ending dates or final extent of individual fire events (Mouillot et al., 2014). This has hampered

distinguishing fire regimes dominated by different fire sizes as both small but frequent fires and large but rare fires may contribute equally to total burned area.

In this sense, global datasets of individual fires derived from pixel-level BA information have recently emerged as an important resource for the fire community, improving our understanding of fire regime (Laurent et al., 2018a). Unlike raw BA products, RS datasets of individual fires provide information beyond the BA, like fire shape, rate of spread and the number of fires (NF).

The Fire Atlas (Andela et al., 2019a, 2019b), FRY (Laurent et al., 2018a, 2018b) and GlobFire (Artés et al., 2019; Artés Vivancos and San-Miguel-Ayanz, 2018) represent the most recent RS individualized fire datasets. These datasets were built from specific algorithms to reconstruct fire patches from MCD64A1 pixel-based BA. In spite of using different methodologies and different assumptions, these datasets shared a common objective: aggregate neighbouring burned pixels with sequential burn dates into individual fire patches.

Though very promising, RS datasets of individual fires have been sparingly compared to historical ground-based fire databases, that are generally thought to be the most reliable source of data regarding fire occurrence and fire extent (Moreira et al., 2011; Mouillot et al., 2014). Previous studies indicated that rigorous evaluation of satellite estimates with ground-based data is needed to assess the reliability of the RS information at regional scale (Turco et al., 2019). Most validation procedures of these RS datasets were based on comparisons between different satellite products (Andela et al., 2019b; Laurent et al., 2018a), with

however scarce attention to independent ground-based observations (Artés et al., 2019).



In this work, we compared the three most recent RS datasets of individual fires (Fire Atlas, FRY and GlobFire) with high-quality fire databases compiled by regional agencies across the most active fire region in Europe (i.e. Southwestern Mediterranean basin) during the common period of observations (2005 to 2015). We sought to provide a solid answer to the following questions. (i) Are RS datasets capturing the actual pattern of fire occurrence and burned area? (ii) To what extent is their accuracy dependent on fire size? (iii) Can we rely on RS datasets to analyze fire regimes? To answer these questions, we assessed the spatial and temporal uncertainty in estimated RS BA and NF aggregated at monthly and 0.25° resolutions across a range of individual fire size thresholds (1 to 500 ha). This study may inform end-users about RS limitations, and provide guidance on the correct usage of global RS information at regional scale.

## 2    Data and Methods

### 2.1    Ground-based fire data

The ground-based dataset was built from multiple regional/national sources, including fire records from Portugal, Spain, France and Sardinia in Italy (Table 1). All these ground monitoring systems provide high-quality datasets that have been extensively used in previous studies across France (Curt et al., 2014), Portugal (Pereira et al., 2011), Sardinia (Salis et al., 2013) and Mediterranean basin (Rodrigues et al., 2020; Turco et al., 2016). Although not free of errors, these datasets constitute the most accurate source of historical information about fires available in Europe.

**Table 1.** Description of regional fire agencies and reference link to the data used to build the ground-based database across Southwest Mediterranean basin.

| Agency | Country | Coverage | Reference link |
|---|---|---|---|
| DECIF | Portugal | National | http://www2.icnf.pt/portal/florestas/dfci/inc/estat-sgif (last access: 10 January 2020) |
| EGIF | Spain | National | https://www.mapa.gob.es/va/desarrollo-rural/estadisticas/Incendios_default.aspx (last access: 18 December 2019) |
| Prométhée | France | Regional | https://www.promethee.com/ (last access: 16 December 2019) |
| Regione Sardegna | Italy | Regional | http://webgis2.regione.sardegna.it/download/ (last access: 22 January 2020) |

A harmonized database was constructed from the aforementioned fire agencies (AG) datasets. We extracted the following information from each regional datasets: the day of ignition, the fire size, and location of the fire event. To ensure consistency across regions and scales, we analyzed the overlapping recording period among the datasets, i.e., 2005–2015. Small fires (<1 ha) were discarded to ensure the coherence of the analysis since these were not reported systematically by agencies over the


studied period. The harmonized database contained 95,561 fire records, including only events that required a firefighting
response (i.e., disregarding agricultural and prescribed fires) (see Fig. 1).

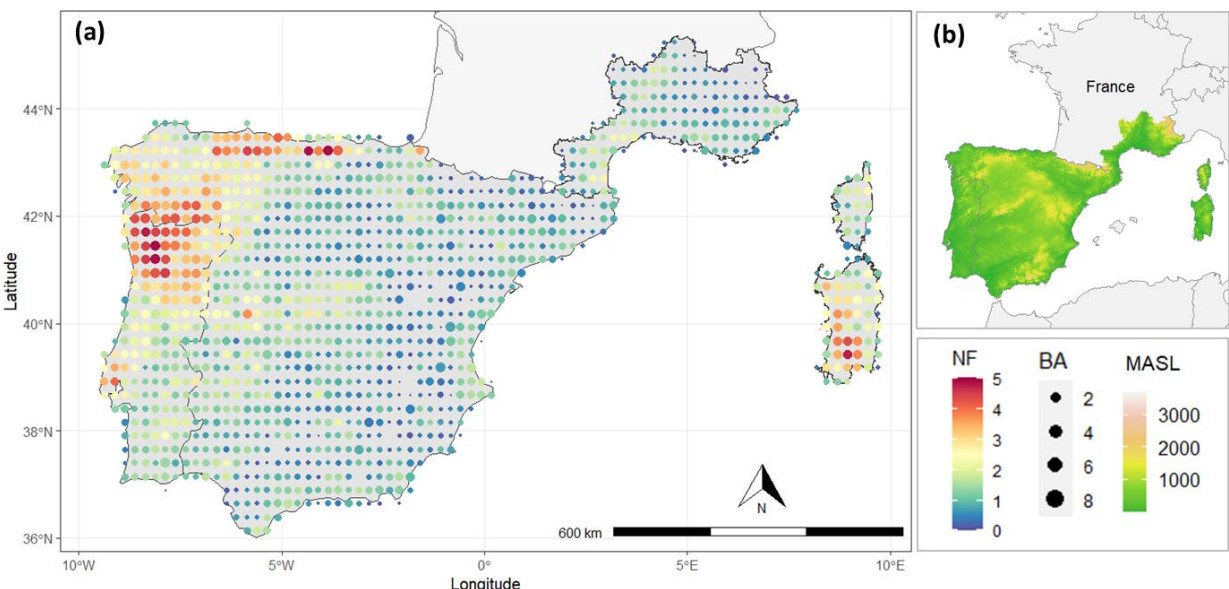

**Figure 1.** (a) Mean annual burned area (BA, depicted by circle size) and mean annual number of fires (NF, depicted by color) at 0.25°
resolution over the study period (2005-2015). Note the logarithmic scale. (b) Orography (in meters).

**2.2    Remotely-sensed fire data**

We used the most recent RS datasets of individual fires: Fire Atlas (Andela et al., 2019a, 2019b), FRY (Laurent et al., 2018a,
2018b) and GlobFire (Artés et al., 2019; Artés Vivancos and San-Miguel-Ayanz, 2018). These datasets provide the date and
the spatial extent of individual fires from the pixel-based burned area MODIS product MCD64A1 Collection 6 (Table 2). The
Terra and Aqua combined MCD64A1 is derived from the surface reflectance imagery and active fires observation. It provides
a global coverage of burned area estimation at a resolution of 500 m (Giglio et al., 2018). The RS datasets of individual fires
were derived using different algorithms such as a progression-based algorithm (Andela et al., 2019), a flood-fill algorithm
(Laurent et al., 2018), and data mining (Artés et al., 2019) that share a common objective: assemble burned pixels that were
adjacent in both space and time to identify and outline individual fire events. All RS dataset provide fire starting and ending
dates, location and the final burned area for each retrieved fire event.
A key parameter of these algorithms is the cut-off value, which is defined as the maximum burn date difference allowed
between two neighbouring pixels to be considered as belonging to the same fire event. This cut-off influences the size, shape
and the degree of clumpiness and fragmentation of individual fire events (Laurent et al., 2018a; Oom et al., 2016). Fire Atlas
used spatially varying cut-off thresholds (4 to 10 days) depending on the fire frequency (Andela et al., 2019b), while the FRY
algorithm processed four different cut-off scenarios (3, 5, 9 and 14 days), used in previous studies (Archibald and Roy, 2009;





Hantson et al., 2015; Nogueira et al., 2017). Finally, GlobFire defined a fire event as a set of burned pixels that are connected within a 5-day window and have not been burned over the 16 previous days (Artés et al., 2019). For simplicity, we only reported the FRY cut-off value that performed the best (5 days). The comparison with all FRY cut-off values is available in the Supplementary material (Fig S1).

**Table 2.** Description of the remote-sensing (RS) datasets of individual fires, including the digital object identifier (DOI) and reference of each dataset. FA: Fire Atlas; FRY_M05: FRY MODIS (5 days) and GF: GlobFire.

| RS dataset | Methodology | Cut-off values | Period | Dataset DOI | Reference |
|---|---|---|---|---|---|
| FA | Progression-based algorithm | 4 to 10 days | 2003-2016 | https://doi.org/10.3334/ORNLDAAC/1642 | (Andela et al., 2019a, 2019b) |
| FRY_M05 | Flood-fill algorithm | 5 days | 2000-2017 | https://doi.org/10.15148/0e999ffc-e220-41ac-ac85-76e92ecd0320 | (Laurent et al., 2018a, 2018b) |
| GF | Data mining | 5 and 16 days | 2000-2019 | https://doi.org/10.1594/PANGAEA.895835 | (Artés et al., 2019; Artés Vivancos and San-Miguel-Ayanz, 2018) |

## 2.3 Methodology

We compared burned area (BA) and number of fires (NF) estimated by RS datasets of individual fires, with the reference ground-based dataset (AG; Fig. 2). We evaluated the ability of RS estimates to reproduce observed temporal and spatial patterns of fire activity observed in AG by fitting ordinary least squares (OLS) linear regressions and using different metrics (OLS slope, R-squared correlation, and bias) to measure RS accuracy. Only the common period between RS datasets and AG records has been considered in the following (2005–2015).

We applied a land cover filter to the RS data using CORINE Land Cover (CLC) 2006 and 2012 to exclude fires located within agricultural or artificial lands that are not always reported by fire agencies. Sensitivity analysis to the land-cover filter is shown in the Supplementary material (Fig S2).

As RS datasets are prone to omit smaller fires due to the coarse spatial resolution of MODIS and other limitations, we investigated different fire size thresholds increasing from 1 to 500 ha. Analyses were repeated for each size-filtered sample (i.e. excluding fires smaller than a given threshold).





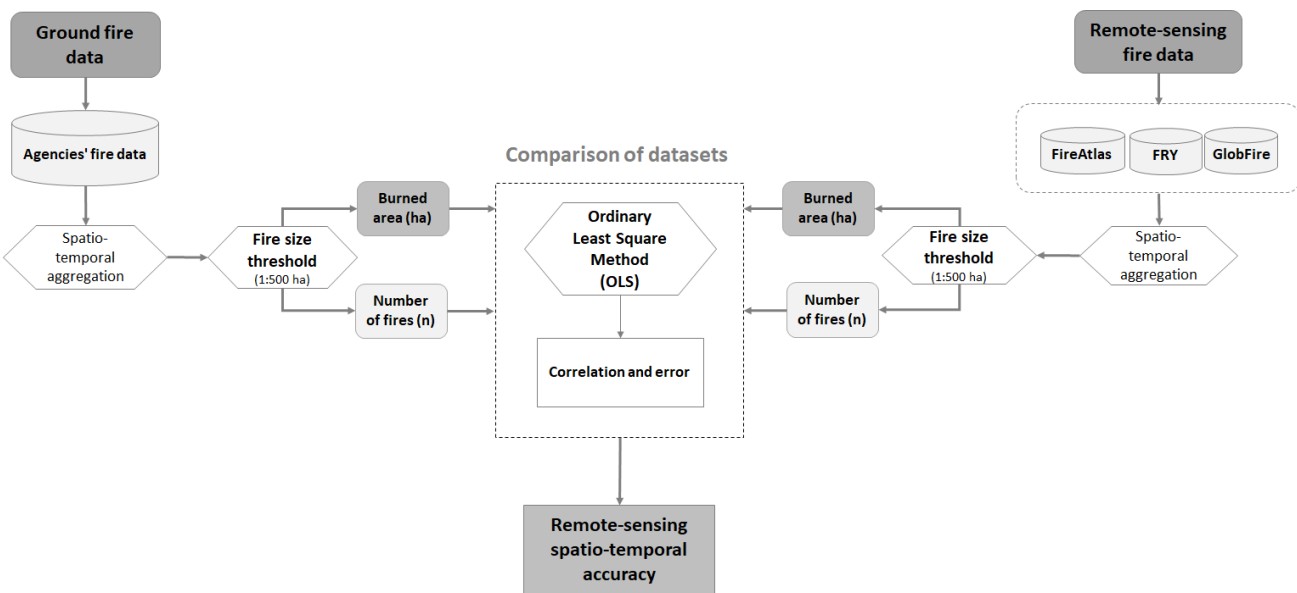

**Figure 2.** The general framework for comparison of RS estimated burned area and number of fires with AG ground-based observations across a range of individual fire size thresholds (1 to 500 ha).

### 2.3.1 Temporal agreement

All datasets were aggregated to monthly scale over the whole study area. We retrieved the slope coefficient of OLS regressions and the coefficient of determination (R-squared) as a proxy of agreement between RS and AG. Slope values greater than 1 indicated an underestimation of fire activity as seen by AG datasets and vice versa. A slope equal to 1 would imply perfect agreement. We also calculated the relative error (ε) as:

$$\varepsilon = 100 \times \frac{RS - AG}{AG} \tag{1}$$

where, *RS* is the total BA or NF detected by remote-sensing datasets and *AG* is the BA or NF reported by the agencies over the study period.

### 2.3.2 Spatial agreement

There is much uncertainty in estimating the ignition point from satellite data, mainly due to the spatial and temporal proximity of fire pixels and the possibility of multiple ignition points in a single fire event (Benali et al., 2016). Likewise, AG databases do not provide systematically ignition points. Thus, to overcome this limitation, we aggregated both RS and AG datasets onto a 0.25° grid (≈ 25 km), setting a common ground for both datasets.





To examine the spatial agreement between RS and AG, we calculated the relative error (Eq. 1) for each grid cell. Finally, we
estimated the overall spatial error, computed as the ε averaged across all grid cells for each dataset.

## 3    Results

### 3.1    Temporal agreement

We first analyzed the monthly distributions of RS and AG observations for all fires (>1 ha) aggregated across the whole studied

area. Fig. 3 shows that RS estimates follow a similar variability in terms of monthly BA but systematically underestimate the
NF with respect to AG. The best agreement between RS estimates and AG occurs mainly during the warm season (May to
October; see Fig. 4). This is usually the period experiencing the largest fires, which account for the bulk of BA in the region
(Turco et al., 2016). Conversely, the poorest agreement was found during the cool season (November to April), a period
dominated mainly by small fires linked to agricultural activities.

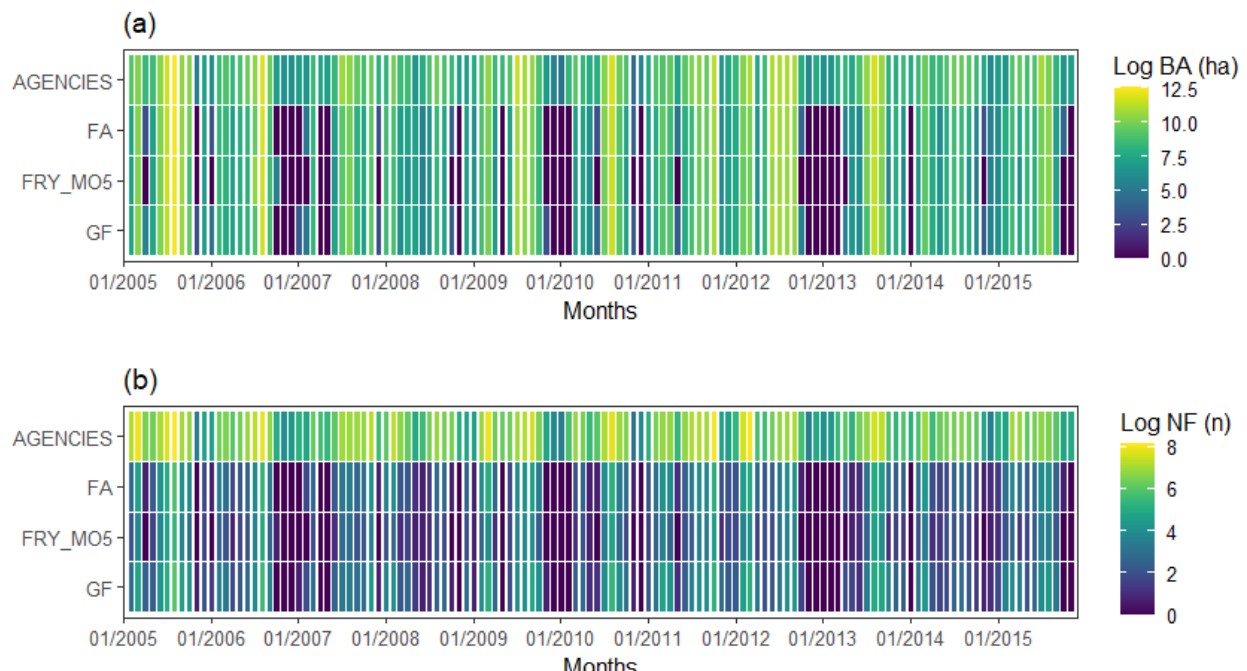

**Figure 3.** (a) Monthly burned area and (b) number of fires (>1 ha) in each fire dataset across the Southwest Mediterranean basin over 2005-2015.

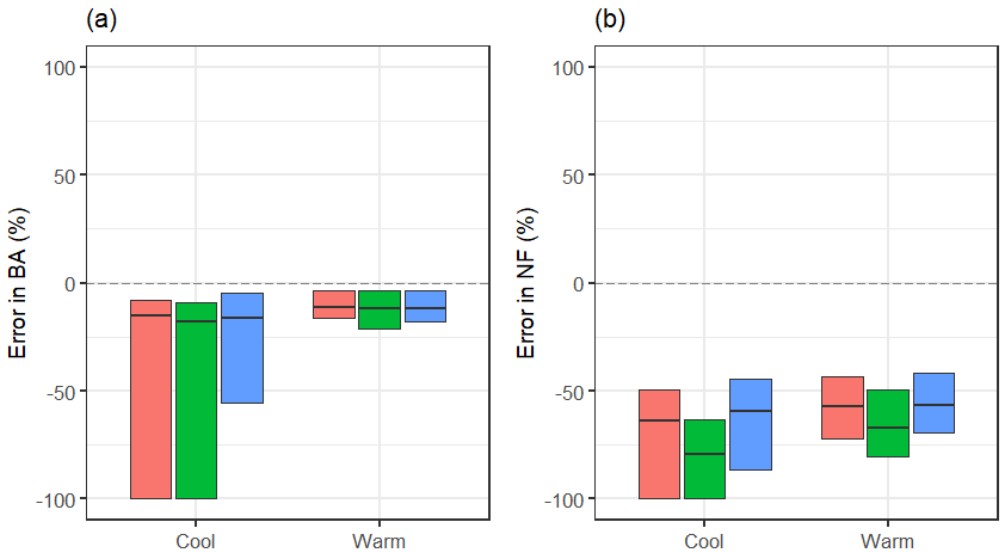

**Figure 4.** (a) Seasonal monthly error (ε) for burned area and (b) number of fires estimates of each RS dataset for all fires >1 ha in the studied area. Cool season from November to April and warm season from May to October. Dashed lines represent the perfect agreement between the datasets.

Table 3 presents the total BA and NF as well as monthly and annual correlation for all fires (>1 ha). Monthly correlations showed a stronger agreement for BA ($R^2 \approx 0.98$) than for NF ($R^2 \approx 0.89$). Annual correlations, where the effect of the seasonal cycle was removed, also showed very high values ($R^2 \approx 0.99$). Despite the fact that RS underestimated the total BA by 38% and the NF by 96% for all fires, they reproduced almost perfectly the temporal variability on both monthly and annual basis. The difference in absolute numbers thus relates to undetected small fires in RS datasets.

**Table 3.** Temporal correlation of monthly and annual burned area and number of fires between RS and AG datasets for all fires (>1 ha) between 2005 and 2015 across the study domain.

| Dataset | Burned area | | | Number of fires | | |
|---|---|---|---|---|---|---|
| | Total (ha) | Mo. correlation | Yr. correlation | Total (n) | Mo. correlation | Yr. correlation |
| AGENCIES | 2,527,603 | - | - | 95,561 | - | - |
| FA | 1,609,267 | 0.99 | 0.99 | 3,875 | 0.90 | 0.99 |
| FRY_M05 | 1,524,171 | 0.99 | 0.99 | 2,134 | 0.88 | 0.99 |
| GF | 1,562,001 | 0.98 | 0.99 | 4,637 | 0.90 | 0.99 |

The monthly agreement of BA and NF (Fig. 5) varies with fire size thresholds (1, 50, 100 and 500 ha). The positive slope of the linear trends indicates that RS generally underestimate both BA and NF when accounting for all fires (> 1 ha). However,

they become progressively more accurate as the fire size threshold increases, a feature that is particularly evident in NF estimates (Fig. 5 e-h).

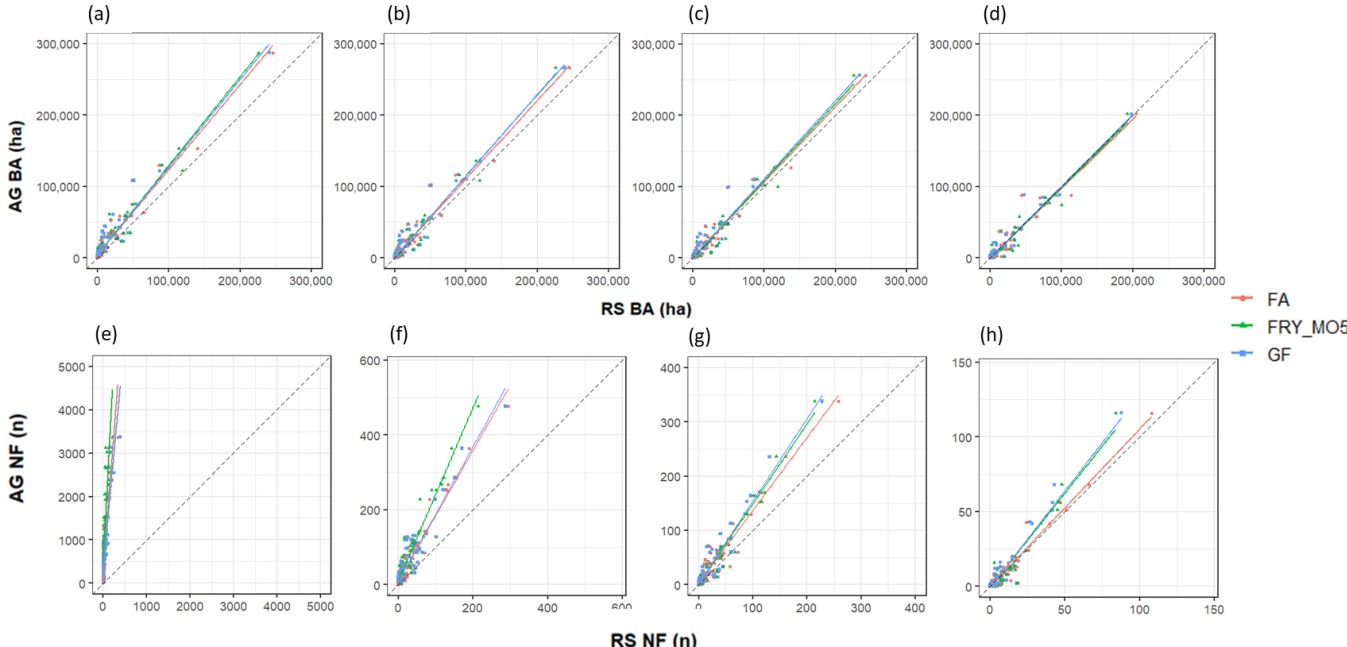


**Figure 5.** Comparison of AG and RS monthly burned area (top) and the number of fires (bottom) when considering a) all fires (> 1 ha), b) fires >50 ha, fires >100 ha and d) fires >500 ha. (e-h) Same as a-d) but for the number of fires. The 1:1 dashed lines represent the perfect fit between the datasets.

Fig. 6 shows the evaluation of RS datasets through different metrics over the continuum of fire size thresholds. Except for the
R-Squared (Fig. 6, middle) which saturates for fires >100 ha for NF, all metrics present a similar behavior showing better agreement when increasing the fire size threshold. Generally, BA (Fig. 6, top) presented better accuracy than NF (Fig. 6, bottom). Despite the different methodologies used to reconstruct individual fires, all datasets followed similar scores, albeit FA displayed lower relative error ($\varepsilon$) for NF.

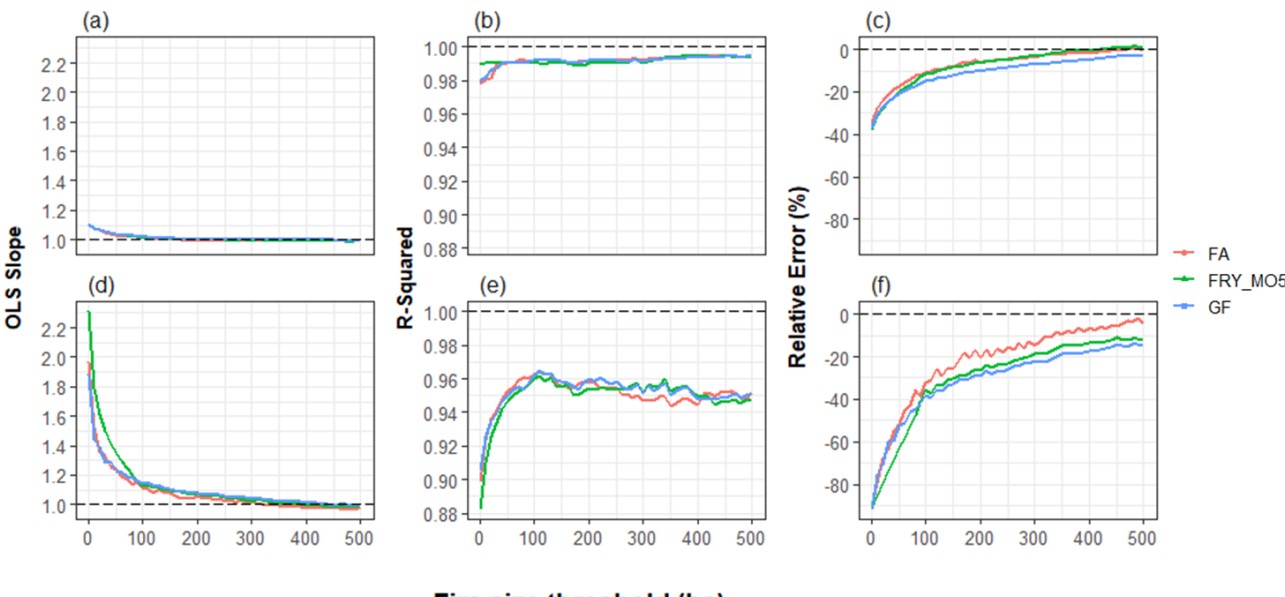

**Figure 6.** Evaluation of RS datasets through different metrics including the slope (left), R-squared correlation (middle) and relative error (right) for both burned area (top) and the number of fires (bottom) over a range of individual fire size thresholds (1 to 500 ha). Dashed lines indicate a perfect fit between RS and AG fire data.

## 3.2 Spatial agreement

Fig. 7 shows the spatial distribution of the relative error ($\varepsilon$) for BA over different individual fire size thresholds (for all fire size thresholds see Supplementary Data). As expected from previous results, RS datasets strongly underestimated BA, especially when including smaller fires. However, a few exceptions are seen for fires < 50 ha mainly over eastern Spain, suggesting that RS detect in that case more fires than AG. This may be related to a few small prescribed fires that are not reported in AG. Also, we found much lower relative errors in regions with higher fire activity, such as the Northern Iberian Peninsula. This is rather expected, as an absolute change in regions with high (low) baseline will result into a small (large) percentage change.

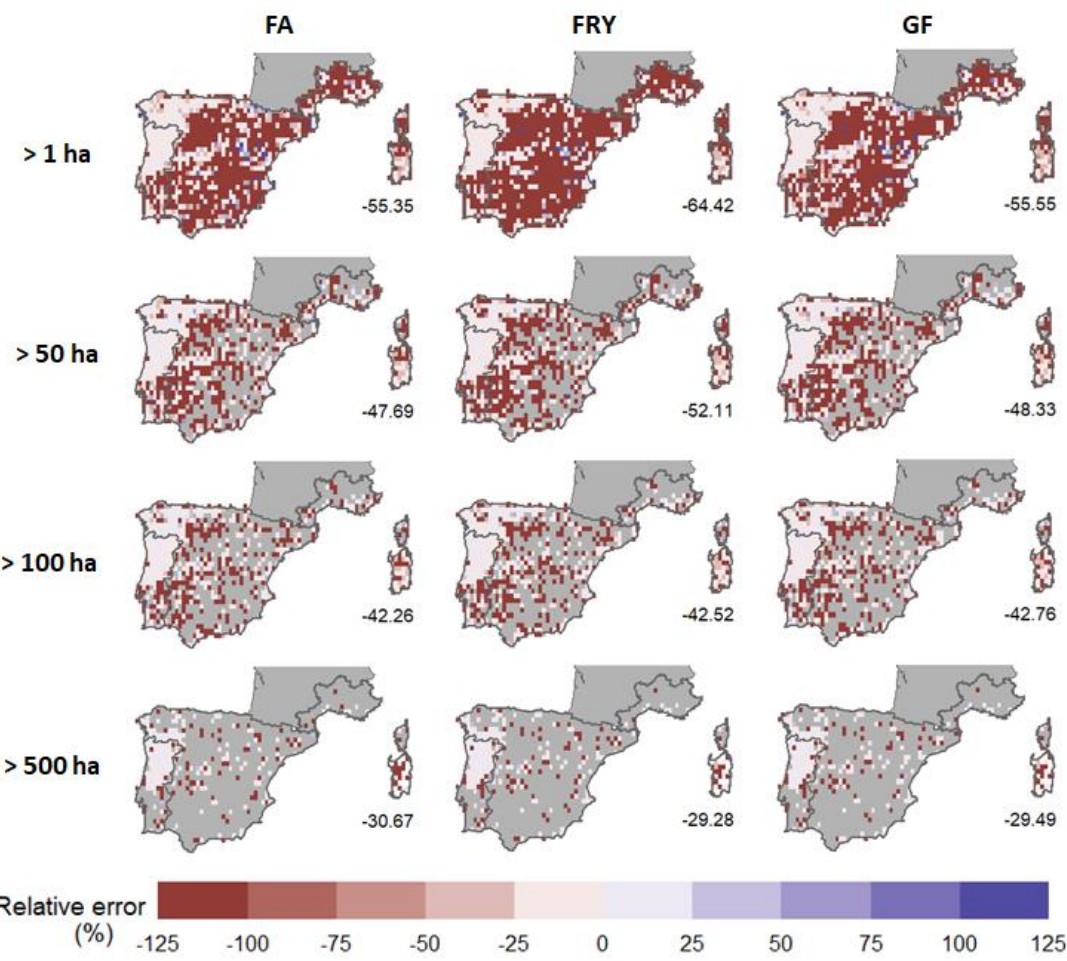

**Figure 7.** The relative error (ε) of the total burned area computed as the relative difference between RS and AG data over different individual fire size thresholds (1, 50, 100 and 500 ha). The overall ε is indicated on each map.

Likewise, RS strongly underestimated NF (Fig. 8), likely disregarding those smaller fires not detected by MODIS.
Surprisingly, a few areas showed positive differences in NF for fires >100 ha across parts of Spain. This overestimation of large fires may be related to the fact that RS algorithms are likely to split larger fires into multiple events. Nevertheless, the overall relative error between RS and AG decreases when focussing on larger fires for both NF and BA, highlighting the important role of fire size on RS accuracy.

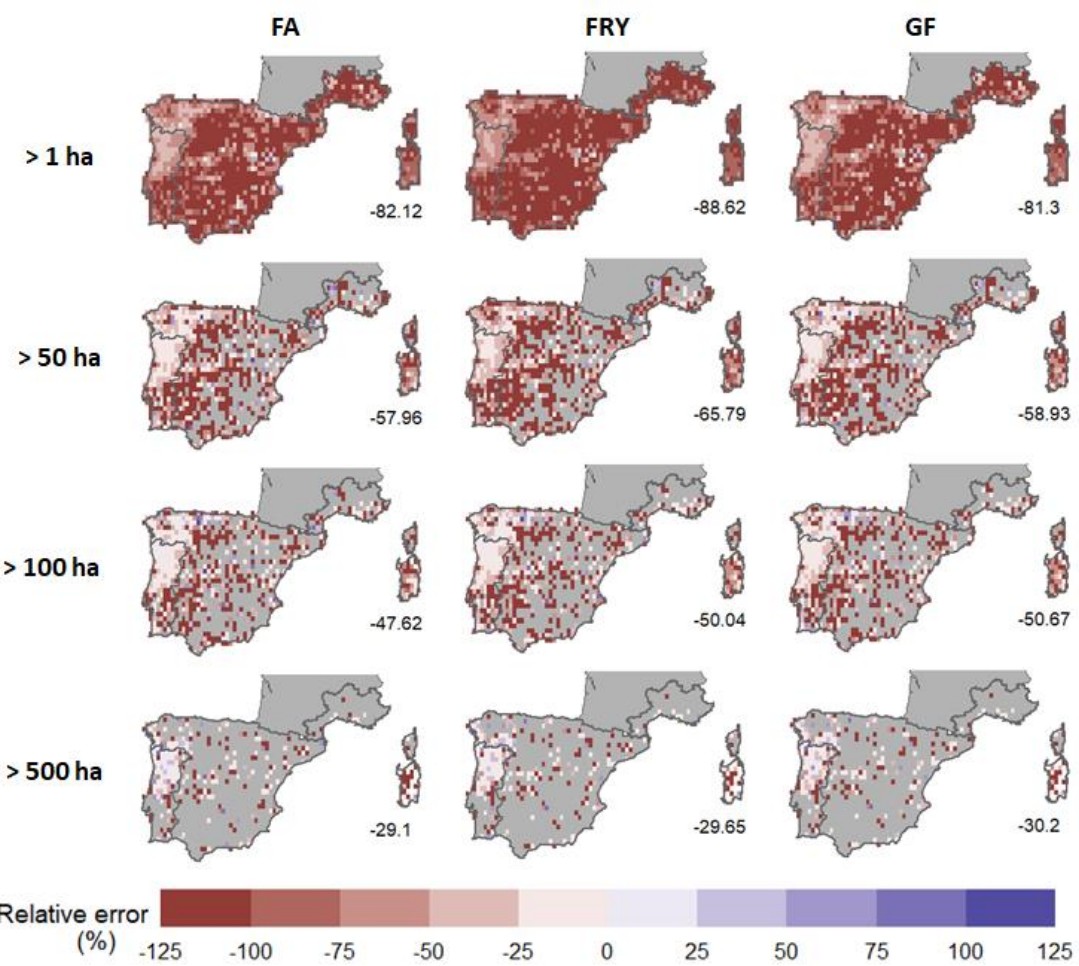

**Figure 8.** Same as Fig. 7 but for NF.

## 4 Discussion

The necessity to properly understand global changes in fire activity calls for efficient and harmonized approaches to record fire incidence. Satellite-borne spectral and thermal sensors offer several global fire products, evolving from BA mapping and active fire detection to novel developments post-processing BA products into single fire datasets (Chuvieco et al., 2019). The ongoing challenge lies in determining their reliability and usefulness. Here, we compared RS with ground-based datasets across the Southwestern Mediterranean basin to better understand RS datasets limitations and guide end-users.

Our results demonstrate that individual fire size plays a major role in the fire detection from RS. Focusing on larger fires (fire typically > 100 ha), RS datasets were in a stronger agreement with AG regardless of the evaluated metrics. Fires > 100 ha denoted much lower error (BA 10%; NF 35%). Likewise, larger fires denoted higher spatial coherence. As expected, the error was lower in areas with higher fire activity such as the northwest of the Iberian Peninsula or the south of Sardinia (Fig.7-



Fig.8). Our findings are in agreement with previous studies, which pointed at fire size as the primary limiting factor for RS estimates (Rodrigues et al., 2019; Ying et al., 2019; Zhu et al., 2017).

The ability of RS datasets to identify individual fires depends mainly on two features: the processing algorithm and the underlying reliability of the BA product. The relatively low capacity of the latter to detect small fires is related to the coarse

spatial resolution (500 m) of the MODIS sensor. Several recent studies have shown that MODIS products rather reliably detect fires over 40–120 ha but miss a number of smaller fires (Fusco et al., 2019; Giglio et al., 2018; Rodrigues et al., 2019; Zhu et al., 2017). Although other BA products, such as FireCCI50 (Chuvieco et al., 2018), provide finer spatial resolution (250 m), a substantial number of small and/or highly fragmented fires remain undetected, leading to a considerable underestimation of BA (Roteta et al., 2019). In addition, all space-borne BA products face many other well-documented limitations such as the

variability in orbital coverage, satellite overpass time, and satellite view obstruction (Cardoso et al., 2005; Padilla et al., 2014). In this sense, detectability may vary regionally across the globe and without ground-based fire datasets, it may be difficult to properly validate their reliability (Turco et al., 2019). Nonetheless, the limitations of MCD64A1 are inherent to all RS dataset, since all of the analyzed products were built on this basis. Hence, differences among RS datasets are rather expected to be associated with the underlying algorithm used to identify single fire events.

RS datasets were found to better simulate BA than NF. This disparity relies on the complexity of extracting individual fires from BA products, including factors that may influence the sensor detection power, resulting in a break in BA continuity thereby increasing the risk of artificially splitting single fires into different fire events. Likewise, if a fire lasts longer than the defined cut-off window, it will be automatically split into different events (Oom et al., 2016). In addition, if multiple fires occur simultaneously in the same region, the parameterization of the RS algorithms may merge multiple individual fires

(Archibald et al., 2013). Lastly, regional features of the fire regime may constrain RS accuracy. For instance, the Mediterranean fire regime is known for hosting numerous small fires, which are unlikely to be detected by RS. These fires do not contribute very much to the total annual burned area but significantly harm the performance of the RS datasets in terms of NF (Turco et al., 2016).

Even though the selection of an appropriate fire size threshold depends on the objectives of each analysis, we can generally

recommend a minimum size of 100 ha, which outstands as a change point in multiple statistics (Fig.6 to Fig.8), with the relative error sharply (dowdily) decreasing in both BA and NF above this threshold. Among the analyzed RS datasets, FA displayed a slightly better performance, with a lower relative error. This may arises from the use of a spatially explicit cut-off threshold, taking both fire spread rate and satellite coverage into account to track the extent of individual fires (Andela et al., 2019b). However, uncertainty in MODIS largely outpaces the uncertainties across the RS datasets.

The spatio-temporal aggregation applied in our study is expected to increase the signal-to-noise ratio and thus decrease the uncertainty in RS estimates. According to Turco (2019), the spatial agreement between AG and RS increases at lower resolutions, being generally best when aggregating the data onto a 1° grid (approximately 110 km) or beyond. Likewise, aggregating the data over time (either monthly or annually) also increases the signal-to-noise ratio by filtering out the temporal stochastic noise (Spadavecchia and Williams, 2009). Evaluating RS datasets on shorter timescales and/or finer spatial



resolutions would likely deteriorate the agreement with AG. Nevertheless, a spatio-temporal aggregation, such as the one employed here, has been extensively used in previous studies analyzing fire regimes at regional (Barbero et al., 2014; Jiménez-ruano et al., 2020; Parisien et al., 2014) and global scales (Bedia et al., 2015; Di Giuseppe et al., 2016; Turco et al., 2018b). Further studies are still needed to examine RS estimates at the fire patch level (i.e. assign individual fires from RS to AG) in order to more precisely quantify RS accuracy at the fire scale.

## 5    Data availability

The above described fire datasets, their characteristics and reference to access the data can be found in Tables 1 and 2. All these fire datasets are open access except one of the ground-based datasets (EGIF) that is available upon request. The different data producers host the data in different ways, typically using websites or data repositories. The harmonized AG database used here as ground-based reference is available at https://doi.org/10.5281/zenodo.3905040 (Galizia et al., 2020).

## 6    Conclusion

In this work, we built upon previous research and investigated the reliability of three RS datasets of individual fires over a range of fire size thresholds across the Southwestern Mediterranean basin. Overall, RS datasets were able to capture reasonably well the temporal and spatial patterns of fire activity, with however limited ability to outline small-to-mid fire events. Despite the different methodologies used to reconstruct fire patches, all datasets (FA, FRY and GlobFire) performed similarly and were increasingly accurate when focusing on larger fires. Specifically, when considering fires > 100 ha, RS denoted reasonable agreement with observed AG data.

Generally, the RS underestimation of BA and NF for smaller fires is related to the coarse spatial resolution (500 m) of the pixel-based BA product and other observation limitations, preventing the detection of small fires. Features of fire regime at regional scales may also influence the RS accuracy (e.g. fire size, density, and spread rate). In this sense, our analysis was framed in the Mediterranean region to capture homogeneous conditions in terms of fire regimes, even though local signals do exist.

We found a better agreement during the warm season (May to October), the main fire season in Southern Europe, especially in regions with higher fire activity (Northern Iberian Peninsula and Southern Sardinia). Also, RS were found to better estimate BA than NF. This is rather expected as numerous small fires, which are not detected by satellites, do not contribute very much to the total burned area across the study region.

Our results may provide guidance for end-users. A quantitative estimate of uncertainty is crucial to the correct interpretation of RS datasets and users should take into account their limitations. Our findings suggested that global RS datasets of individual fires can be used for fire modeling, however caution is advised when drawing from smaller fires (< 100 ha) across the



Mediterranean region. Future studies using high-quality ground-based fire data in other regions of the world featuring different
fire regimes would provide further insights on RS uncertainties.

**Author contributions.** LG carried out the analysis. All authors contributed to the design of the methodology, to discuss the results and to writing the paper.

**Competing interests.** The authors declare that they have no conflict of interest.

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
