# Peer review of "Assessing the accuracy of remotely-sensed fire datasets across the Southwestern Mediterranean basin"

_Natural Hazards and Earth System Sciences, 2020_

## Referee Comment (RC1) · Anonymous Referee #1 · 5 Aug 2020

The paper describes a comparative analysis of the performance of remotely sensed fire datasets. This work is quite interesting and could lead to important discoveries in future forest fire work.

I believe your paper would benefit greatly from some minor corrections. Specifically, I would like to see some improvements on the overall flow of the paper. Lines 45, 62, 64, 71 and 247 seem to require technical corrections as do some other areas.

Figure 1 requires some work as it is unclear what units are being used for the legends and MASL is not explained or found anywhere else in the text.

Figure 4 requires a legend.

[Figure]

I feel these corrections would make this article an excellent research article suitable for publication.

---

## Author Comment (AC1) · 14 Aug 2020

General comments:

The paper describes a comparative analysis of the performance of remotely sensed fire datasets. This work is quite interesting and could lead to important discoveries in future forest fire work. I believe your paper would benefit greatly from some minor corrections. Specifically, I would like to see some improvements on the overall flow of the paper. I feel these corrections would make this article an excellent research article suitable for publication.

We thank the reviewer for the positive comments to our manuscript.

Specific comments:

Lines 45, 62, 64, 71 and 247 seem to require technical corrections as do some other areas.

L 45: We rewrote the sentence as "The MODIS sensor outstands as one of the best data provider for most burned area products such as MCD64A1 (Giglio et al., 2018) and FireCCI50 (Chuvieco et al., 2018)."

L 62: We rewrote the sentence as "Previous studies indicated that rigorous evaluation of satellite data with ground-based data is needed (Turco et al., 2019)."

L 71: We rewrote the sentence as "To answer these questions, we assessed both spatial and temporal uncertainties in RS BA and NF data aggregated at monthly and 0.25° resolutions across a range of individual fire size thresholds (1 to 500 ha)."

L 247: We corrected the sentence as "This may arise from the use of a spatially explicit cut-off threshold…"

Figure 1 requires some work as it is unclear what units are being used for the legends and MASL is not explained or found anywhere else in the text.

In response to this suggestion, Figure 1 was modified and we replaced the topography with the countries' boundaries to guide the reader. Please see below:

[Figure]

**Figure 1.** (a) Mean annual burned area (BA, depicted by circle size) and mean annual number of fires (NF, depicted by color) at 0.25° resolution over the study period (2005-2015). (b) Spatial extent of the AG dataset.

Figure 4 requires a legend.

The legend of Figure 4 has been included. Please see below:

[Figure]

**Figure 4.** Seasonal error ($\varepsilon$) for (a) burned area and (b) number of fires in each RS dataset for all fires >1 ha across the studied area with respect to AG dataset. Cool season from November to April and warm season from May to October.

---

## Referee Comment (RC2) · Anonymous Referee #2 · 19 Sep 2020

This paper presents a comparative assessment analysis of three primary fire datasets across the Southwestern Mediterranean region, including Spain, Portugal, Frace, and Italy. The paper, its structure, and its analyses have been well designed and presented. The paper was also easy to read. Nevertheless, some minor issues can be addressed to improve the technical and presentation quality of the paper. There are as follows: - I think the motivation and the contribution of the paper still are not well presented. So, I invite the authors to clearly explained the novelty of their research and its applications and advantages for the real-work uses. - In the abstract, the abbreviation AG has been defined for "fire agencies," but later in a sentence: "Our results show that RS datasets were highly correlated with AG." It is unclear to the reader what the authors do mean

here. I would suggest they review these sentences from lines 14 to 17. - The same problem is there again for the sentence 19-20: How RS (Remote Sensing) and AG (fire agencies) can be in agreement! - In the line 128, the authors have mentioned that investigated different fire size thresholds increasing from 1 to 500 ha. It would be interesting for the reader to know how it has been applied to the original dataset from MODIS that does have a spatial resolution of 500 m. - Again, in equation (1) in section 2.3.1, the authors have used AG as mathematical/physical parameters. However, it was merely an abbreviation for Fire Agency. More clarification here would be helpful for the readers. - The quality of some graphics and figures can be increased to show more details, e.g., Fig. 7 & 8 are interesting, but the details are not visible. - A think the figures in the supplementary material can also be added to the main manuscript. They will help the readers to see all the information related to study in the same way. - There some long and complex sentences across the manuscript that do the reading and understanding of the text challenging. As a result, final proofreading would help solve these issues.

---

## Author Comment (AC2) · 8 Oct 2020

General comments:

This paper presents a comparative assessment analysis of three primary fire datasets across the Southwestern Mediterranean region, including Spain, Portugal, France, and Italy. The paper, its structure, and its analyses have been well designed and presented. The paper was also easy to read. Nevertheless, some minor issues can be addressed to improve the technical and presentation quality of the paper.

We thank the reviewer for the positive comments to our manuscript. We followed the reviewer's suggestion and modified some parts of the manuscript to improve it.

Specific comments:

I think the motivation and the contribution of the paper still are not well presented. So, I invite the authors to clearly explained the novelty of their research and its applications and advantages for the real-work uses.

In order to clarify the novelty and application of our work, we included the following sentences in the manuscript. Please see below:

L 66: In this work, we compared for the first time the three most recent remotely-sensed fire datasets of individual fires (Fire Atlas, FRY and GlobFire) with high-quality fire datasets compiled by regional agencies across the most active fire region in Europe (i.e. Southwestern Mediterranean basin) during the common period of observations (2005 to 2015).

L 68: While most previous studies have evaluated remotely-sensed data on a fire-by-fire basis, this study aggregates individual fires across months and pixels (0.25°) and seeks to estimate how much temporal variability in both fire frequency and burned area are captured by remotely-sensed datasets.

L 250: The low capacity of gridded BA products to detect small-mid fire events (< 100 ha) can be improved by the generation of products based on higher resolution sensors in the range of 10–30m (Roteta et al., 2019). RSD of individual fires derived from finer gridded BA would provide better accuracy in the fire metrics, specifically for NF. In addition, the MCD64A1 product already incorporates the uncertainty of detection as an auxiliary variable of gridded BA data (Giglio et al., 2018). RSD could benefit from this and report similar information at individual fire level.

L 281: In practical applications, our results may provide guidance for end-users regarding RSD limitations at different fire size thresholds.

L 284: Fire agencies may benefit from the spatial and temporal consistency of remotely sensed data to support their operational fire mapping system at regional/national level.

In the abstract, the abbreviation AG has been defined for "fire agencies," but later in a sentence: "Our results show that RS datasets were highly correlated with AG." It is unclear to the reader what the authors do mean here. I would suggest they review these sentences from lines 14 to 17.

In response to this suggestion, we included the abbreviations (i.e. RSD: remote-sensing datasets; AGD: agency dataset) in the manuscript. Please see below:

L 9: Although very promising, these datasets still lack a quantitative estimate of their accuracy with respect to fire agencies (AG) historical records.

L 12: Here, we compared three state-of-the-art RS datasets (RSD; Fire Atlas, FRY and GlobFire) with a harmonized fire agency dataset (AGD) compiled by ground-based monitoring systems across the Southwestern Mediterranean basin (2005-2015).

L 14: We assessed the spatial and temporal accuracy of RSD with respect to both burned area (BA) and number of fires (NF). RSD and AGD were aggregated at monthly and 0.25° resolutions, considering different individual fire size thresholds ranging from 1 to 500 ha. Our results show that both datasets were highly correlated, in terms of monthly BA and NF but RSD severely underestimated both (by 38% and 96%, respectively) when considering all fires > 1 ha.

L 76: The fire agency dataset (AGD) was built from multiple ground-based sources, including records from Portugal, Spain, France and Sardinia in Italy (Table 1).

L 85: A harmonized database was constructed from the aforementioned fire agencies (AG) datasets. (deleted)

L 96: We used the most recent remote-sensing datasets (RSD) of individual fires: Fire Atlas (Andela et al., 2019a, 2019b), FRY (Laurent et al., 2018a, 2018b) and GlobFire (Artés et al., 2019; Artés Vivancos and San-Miguel-Ayanz, 2018).

L: 103: All RSD provide fire starting and ending dates, location and the final burned area for each retrieved fire event.

L 115: Table 2. Description of the remote-sensing datasets (RSD) of individual fires, including the digital object identifier (DOI) and reference of each dataset. FA: Fire Atlas; FRY_M05: FRY MODIS (5 days) and GF: GlobFire.

L 119: We compared burned area (BA) and number of fires (NF) estimated by RSD of individual fires, with the ground-based reference AGD (Fig. 2).

L 120: We evaluated the ability of RSD to reproduce observed temporal and spatial patterns of fire activity observed in AGD by fitting ordinary least squares (OLS) linear regressions and using different metrics (OLS slope, R-squared correlation, and bias) to measure RSD accuracy. Only the common period between RSD datasets and AGD has been considered in the following (2005–2015).

L 124: We applied a land cover filter to the RSD using CORINE Land Cover (CLC) to exclude fires located within agricultural or artificial lands that are not reported by the fire agencies.

L 133: Figure 2. The general framework for comparison of RSD burned area and number of fires with AGD ground-based observations across a range of individual fire size thresholds (1 to 500 ha).

L 136: We retrieved the slope coefficient of OLS regressions and the coefficient of determination (R-squared) as a proxy of agreement between RSD and AGD.

L 137: Slope values greater than indicated an underestimation of fire activity as seen by AGD and vice versa.

L 147: Likewise, AGD databases do not provide systematically ignition points.

L 148: Thus, to overcome this limitation, we aggregated both RSD and AGD datasets onto a 0.25° grid (≈ 25 km), setting a common ground for both datasets.

L 150: To examine the spatial agreement between RSD and AGD, we calculated the relative error (Eq. 1) for each grid cell.

L 154: We first analyzed the monthly distributions of RSD and AGD for all fires (>1 ha) aggregated across the whole studied area.

L 156: The best agreement between RSD and AGD occurs mainly during the warm season (May to October; see Fig. 4).

L 172: Table 3. Temporal correlation of monthly and annual burned area and number of fires between RSD and AGD for all fires (>1 ha) between 2005 and 2015 across the study domain.

L 184: Fig. 6 shows the evaluation of RSD through different metrics over the continuum of fire size thresholds.

L 190: Figure 6. Evaluation of RSD through different metrics including the slope (left), R-squared correlation (middle) 190 and relative error (right) for both burned area (top) and the number of fires (bottom) over a range of individual fire size thresholds (1 to 500 ha).

L 195: As expected from previous results, RSD strongly underestimated BA, especially when including smaller fires. However, a few exceptions are seen for fires < 50 ha mainly over eastern Spain, suggesting that RSD detect in that case more fires than AGD. This may be related to a few small prescribed fires that are not reported in AGD.

L 204: Likewise, RSD strongly underestimated NF (Fig. 8), likely disregarding those smaller fires not detected by MODIS.

L 206: Nevertheless, the overall relative error between RSD and AGD decreases when focusing on larger fires for both NF and BA, highlighting the important role of fire size on RSD accuracy.

L 215: Here, we compared remotely-sensed fire data with ground-based datasets across the Southwestern Mediterranean basin to better understand RSD limitations and guide end-users.

L 223: The ability of RSD to determine individual fires depends mainly on two features: the processing algorithm and the underlying reliability of the BA product.

L 233: Hence, differences among RSD are rather expected to be associated with the underlying algorithm used to identify single fire events.

L 235: RSD were found to better estimate BA than NF.

L 240: Lastly, regional features of the fire regime may constrain RSD accuracy.

L 241: These fires do not contribute very much to the total annual burned area but significantly harm the performance of the RSD estimations in terms of NF (Turco et al., 2016).

L 246: Among the analyzed RSD datasets, FA displayed a slightly better performance, with a lower relative error.

L 249: However, uncertainty in MODIS estimations largely outpaces the uncertainties across the RSD.

L 251: According to Turco (2019), the spatial agreement between AGD and RSD increases at lower resolutions, being generally best when aggregating the data onto a 1° grid (approximately 110 km) or beyond.

L 254: Evaluating RSD on shorter timescales and/or finer spatial resolutions would likely deteriorate the agreement with AGD.

L 258: Further studies are still needed to examine RSD spatio-temporal variability at the fire patch level (i.e. assign individual fires from RSD to AGD) in order to more precisely quantify the dataset accuracy at the fire scale.

L 270: Specifically, when considering fires > 100 ha, RSD denoted reasonable agreement with observed AGD.

L 272: Generally, the RSD underestimation of BA and NF for smaller fires is related to the coarse spatial resolution (500 m) of the pixel-based BA product and other observation limitations, preventing the detection of small fires. Features of fire regime at regional scales may also influence the RSD accuracy (e.g. fire size, density, and spread rate).

L 278: Also, RSD were found to better estimate BA than NF. This is rather expected as numerous small fires, which are not detected by satellites, do not contribute very much to the total burned area across the study region.

L 281: A quantitative estimate of uncertainty is crucial to the correct interpretation of RSD and users should take into account their limitations.

L 282: Our findings suggested that global RSD of individual fires can be used for fire modeling, however caution is advised when drawing from smaller fires (< 100 ha) across the Mediterranean region. Future studies using high-quality ground-based fire data in other regions of the world featuring different fire regimes would provide further insights on RSD uncertainties.

The same problem is there again for the sentence 19-20: How RS (Remote Sensing) and AG (fire agencies) can be in agreement!

In response to this suggestion, we modified the following sentences. Please see below:

L 17: The agreement between RSD and AGD strengthens when increasing the fire size threshold, with fires > 100 ha denoting higher correlation and much lower error (BA 10%; NF 35%).

L 19: The agreement between RSD and AGD was also the highest during the warm season (May to October) in particular across the regions with greater fire activity such as the Northern Iberian Peninsula.

In the line 128, the authors have mentioned that investigated different fire size thresholds increasing from 1 to 500 ha. It would be interesting for the reader to know how it has been applied to the original dataset from MODIS that does have a spatial resolution of 500 m.

In response to this suggestion, we included complementary information in the following sentence. Please see below:

L 127: As RSD are prone to omit smaller fires (<25 ha) due to the coarse spatial resolution of MODIS product MCD64A1 (500 m) and other limitations, we investigated different fire size thresholds increasing from 1 to 500 ha.

Again, in equation (1) in section 2.3.1, the authors have used AG as mathematical/physical parameters. However, it was merely an abbreviation for Fire Agency. More clarification here would be helpful for the readers.

We rewrote and moved the equation to section 2.3.

L 122: We calculated the relative error (ε) as:

$$\varepsilon = 100 \times \frac{BA_{RSD} - BA_{AGD}}{BA_{AGD}} \qquad (1)$$

where, $BA_{RSD}$ represents the burned area (BA) detected by remote-sensing datasets (RSD) and $BA_{AGD}$ represents the burned area registered in the fire agencies datasets (AGD) over the study period. The analysis was repeated for the number of fires (NF).

L 139: We also calculated the relative error (Eq. 1) over the study period.

The quality of some graphics and figures can be increased to show more details, e.g., Fig. 7 & 8 are interesting, but the details are not visible.

We increased the graphical resolution of all Figures of the manuscript. For instance, see below Fig 7 and 8:

[Figure]

**Figure 7.** The relative error ($\varepsilon$) of the total BA computed as the relative difference between $BA_{RSD}$ and $BA_{AGD}$ data over different individual fire size thresholds (1, 50, 100 and 500 ha). The overall $\varepsilon$ is indicated on each map.

[Figure]

**Figure 8.** Same as Fig. 7 but for NF.

I think the figures in the supplementary material can also be added to the main manuscript. They will help the readers to see all the information related to study in the same way.

We agree that the Supplementary Material might help the reader to understand the methodology applied in our study, however, they are not critical to support the conclusion of the manuscript. The supplementary information is an outcome related to the pre-processing of the RS datasets, which is a preliminary step in the methodology section to implement the RS x AG comparison properly. In this sense, we think the inclusion of this information in the main text would disrupt the flow of the manuscript.

We however included the Supplementary material as part of the manuscript in Appendix A (Fig. A1 and A2), which are referred to in the main text (L 113 and L 126). In this sense, complementary information can be easily accessed by the reader and may support the understanding of the applied methodology.

**Appendix A**

[Figure]

**Figure A1.** Evaluation of RSD including all FRY cut-off values (3 to 14 days) through different metrics including the slope (left), R-squared correlation (middle) and relative error (right) for both burned area (top) and the number of fires (bottom) over a range of individual fire size thresholds (1 to 500 ha). Dashed lines indicate a perfect fit between RSD and AGD.

[Figure]

**Figure A2.** Evaluation of "raw" RSD (i.e. without the land cover filter) through different metrics including the slope (left), R-squared correlation (middle) and relative error (right) for both burned area (top) and the number of fires (bottom) over a range of individual fire size thresholds (1 to 500 ha). Dashed lines indicate a perfect fit between RSD and AGD.

There some long and complex sentences across the manuscript that do the reading and understanding of the text challenging. As a result, final proofreading would help solve these issues.

In response to this suggestion, we have rewritten some sentences to improve the readability. Please see below:

L 11: Although very promising, these datasets still lack a quantitative estimate of their accuracy with respect to ground-based fire datasets.

L 22: Overall, our findings suggest a reasonable agreement between RSD and AGD for fires larger than 100 ha, but care is needed when examining smaller fires at regional scales.

L 25: Over the past four decades, there were an average of 47,766 fires annually and an average of 413,209 hectares burned annually (San-Miguel-Ayanz et al., 2017) causing extensive economic and ecological losses, and even human casualties (Keeley et al., 2011; Molina-Terrén et al., 2019).

L 29: The Mediterranean fire regime is dominated by human-caused ignitions (Ganteaume et al., 2013) with most of the total burned area (BA) linked to a limited number of large fires during the summer (Turco et al., 2016).

L 34: Projecting future changes to fire activity requires modeling efforts across broad geographical scales to better understand processes and mechanisms conductive to fire ignition and spread.

L 81: Table 1. Description of regional fire agencies and reference link to the data used to build the ground-based dataset across Southwest Mediterranean basin.

L 89: The harmonized database contained 95,561 fires including only events that required a firefighting response (i.e., disregarding agricultural and prescribed fires) (see Fig. 1).

L 100: Fires were individualized from different algorithms such as a progression-based algorithm (Andela et al., 2019), a flood-fill algorithm (Laurent et al., 2018), and data mining (Artés et al., 2019) that share a common objective: assemble burned pixels that were adjacent in both space and time to identify and outline individual fire.

L 125: To account for the land cover changes over the study period, we used CLC 2006 as a reference to filter RSD from the 2005-2009 period and CLC 2012 from 2010-2015. Sensitivity analysis to the land-cover filter is shown in the Appendix A (Fig. A2).

L 146: We then sought to examine how the agreement between RSD and AGD datasets varies across space.

L 155: Fig. 3 shows that RSD follow a similar variability in terms of monthly BA but systematically underestimate BA and NF with respect to AGD.

L 166: Table 3 presents the total BA and NF as well as monthly (i.e. including the seasonal cycle) and annual correlation (i.e. excluding the seasonal cycle) for all fires (>1 ha).

L 169: Despite the fact that RSD underestimated the total BA by 38% and the NF by 96% for all fires, they reproduced almost perfectly the temporal variability on both monthly and annual basis.

L 176: The positive slope of the linear trends indicates that RSD generally underestimate both BA and NF when accounting for all fires (> 1 ha).

L 217: Our results demonstrate that agreement between RSD and AGD is strongly dependent on individual fire size.

L 235: This disparity relies on the complexity of extracting individual fires from gridded BA products. Environmental conditions (e.g. topography, cloud/smoke cover) may influence the sensor detection power, resulting in a break in BA continuity thereby increasing the risk of artificially splitting single fires into different fire events.

L 244: The selection of an appropriate fire size threshold depends on the objectives of each analysis. However, in this study we can generally recommend a minimum size of 100 ha, which outstands as a change point in multiple statistics (Fig.6 to Fig.8), with the relative error sharply (dowdily) decreasing in both BA and NF above this threshold.

L 264: The harmonized AGD used here as ground-based reference is available at https://doi.org/10.5281/zenodo.3905040 (Galizia et al., 2020).

L 266: In this work, we built upon previous research and investigated the reliability of three RSD of individual fires (FA, FRY and GlobFire) over a range of fire size thresholds across the Southwestern Mediterranean basin.

L 267: Overall, RSD contain only a small fraction of the total number of fires documented by AGD. However, they capture reasonably well the temporal variability of fire activity across monthly and annual scales.

L 269: Despite the different methodologies used to reconstruct fire patches, all RSD performed similarly and were increasingly accurate when focusing on larger fires.

---

## Author Response (AR1)

General comments:

We thank the reviewers for the positive comments to our manuscript. We followed the suggestions and modified some parts of the manuscript to improve it.

In response to the reviewer's suggestions, we harmonized some terms with respect to the analyzed data and included the abbreviations (i.e. RSD: remote-sensing datasets; GBD: ground-based dataset) in the manuscript. In order to clarify the novelty and application of our work, we included some new sentences in the manuscript and we have rewritten others to improve the readability.

We included the Supplementary material as part of the manuscript in Appendix A (Fig. A1 and A2). In this sense, complementary information can be easily accessed by the reader and may support the understanding of the applied methodology. Finally, we also increased the graphical resolution (300 dpi) of all Figures of the manuscript to improve visualization.

Detailed changes in the revised version of the manuscript:

L12-14: Here, we compared three state-of-the-art remote-sensing datasets (RSD; Fire Atlas, FRY and GlobFire) with a harmonized ground-based dataset (GBD) compiled by fire agencies monitoring systems across the Southwestern Mediterranean basin (2005-2015).

L14-15: We assessed the agreement between RSD and GBD with respect to both burned area (BA) and number of fires (NF).

L18-20: The agreement between RSD and GBD was strongly dependent on individual fire size and strengthened when increasing the fire size threshold, with fires > 100 ha denoting higher correlation and much lower error (BA 10%; NF 35%).

L26-28: Vegetation fires are a common and destructive hazard in the Southwestern Mediterranean basin. Over the past four decades, there were, on average, 47,766 fires and 413,209 ha burned annually in this region (San-Miguel-Ayanz et al., 2017) causing extensive economic and ecological losses, and even human casualties (Keeley et al., 2011; Molina-Terrén et al., 2019).

L35-36: Projecting future changes to fire activity requires modeling efforts across broad geographical scales to better understand processes and mechanisms conductive to fire ignition and spread.

L66-68: In this work, we compared for the first time the three most recent remote-sensing datasets of individual fires (Fire Atlas, FRY and GlobFire) with quality-controlled fire databases compiled by regional agencies across the most active fire region in Europe (i.e. Southwestern Mediterranean basin) during the common period of observations (2005 to 2015).

L68-71: While most previous studies have evaluated remote-sensing data on a fire-by-fire basis, this study aggregates individual fires across months and pixels (0.25°) and seeks to estimate to what extent the temporal variability in both fire frequency and burned area are captured by remote-sensing datasets.

L73-74: To answer these questions, we examined the agreement between remotely-sensed and ground-based fire datasets aggregated at monthly and 0.25° resolutions across a range of individual fire size thresholds (1 to 500 ha).

L74-75: This study may inform end-users about remote-sensing datasets' ability to proxy actual fire activity but also on their limitations.

L78-79: The ground-based dataset (GBD) was built from multiple fire agencies sources, including fire records from Portugal, Spain, France and Sardinia in Italy (Table 1).

L84: Table 1. Fire agencies and reference links to the data used to build the harmonized ground-based dataset (GBD) across the Southwest Mediterranean basin.

| Agency | Country | Coverage | Reference link |
|---|---|---|---|
| DECIF | Portugal | National | http://www2.icnf.pt/portal/florestas/dfci/relat/rel-if (last access: 10 January 2020) |
| EGIF | Spain | National | https://www.mapa.gob.es/va/desarrollo-rural/estadisticas/Incendios_default.aspx (last access: 18 December 2019) |
| Prométhée | France | Regional | https://www.promethee.com/ (last access: 16 December 2019) |
| Regione Sardegna | Italy | Regional | http://webgis2.regione.sardegna.it/download/ (last access: 22 January 2020) |

L92:

[Figure]

**Figure 1.** (a) Mean annual burned area (BA, depicted by circle size) and mean annual number of fires (NF, depicted by color) at 0.25° resolution over the study period (2005-2015). (b) Spatial extent of the study area.

L96-97: We used the most recent remote-sensing datasets (RSD) of individual fires: Fire Atlas (Andela et al., 2019a, 2019b), FRY (Laurent et al., 2018a, 2018b) and GlobFire (Artés et al., 2019; Artés Vivancos and San-Miguel-Ayanz, 2018).

L100-102: Fires were individualized from different algorithms such as a progression-based algorithm (Andela et al., 2019), a flood-fill algorithm (Laurent et al., 2018), and data mining (Artés et al., 2019) that share a common objective: assemble burned pixels that were adjacent in both space and time to identify and outline individual fire events.

L112-113: The comparison with all FRY cut-off values is available in the Appendix A (Fig A1).

L119: We compared burned area (BA) and number of fires (NF) estimated by RSD, with the ground-based reference GBD (Fig. 2).

L131-133: To account for the land cover changes over the study period, we used CLC 2006 as a reference to filter RSD from the 2005-2009 period and CLC 2012 from 2010-2015. Sensitivity analysis to the land-cover filter is shown in the Appendix A (Fig. A2).

L134-135: As RSD are prone to omit smaller fires (<25 ha) due to the coarse spatial resolution of MODIS product MCD64A1 (500 m) and other limitations, we investigated different fire size thresholds increasing from 1 to 500 ha.

L148: We then sought to examine how the agreement between RSD and GBD datasets varies across space.

[Figure]

L163:

**Figure 3.** (a) Monthly burned area and (b) number of fires (>1 ha) in each fire dataset across the Southwest Mediterranean basin over 2005-2015.

80   L 166:

[Figure]

**Figure 4.** (a) Median and inter-quartile range of the seasonal error (ε) observed each year for burned area and (b) number of fires estimates of each RSD for all fires >1 ha in the studied area. Cool season from November to April and warm season from May to October. Dashed lines represent the perfect agreement between the datasets.

85   L170-171: Table 3 presents the total BA and NF as well as monthly (i.e. including the seasonal cycle) and annual correlation (i.e. excluding the seasonal cycle) between RSD and GBD for all fires (>1 ha).

L184:

[Figure]

**Figure 5.** Comparison of GBD and RSD in respect to monthly burned area (top) and the number of fires (bottom) when considering a) all fires (> 1 ha), b) fires >50 ha, fires >100 ha and d) fires >500 ha. (e-h) Same as a-d) but for the number of fires. The 1:1 dashed lines represent the perfect fit between the datasets.

L222-223: Although RSD may miss a substantial number of fires, the temporal variations in both NF and BA match very well with ground-based observations.

L223-224: Our results also demonstrate that agreement between RSD and GBD is strongly dependent on individual fire size.

L242-244: Environmental conditions (e.g. topography, cloud/smoke cover) may influence the sensor detection power, resulting in a break in BA continuity thereby increasing the risk of artificially splitting single fires into different fire events.

L270-271: Further studies are still needed to examine RSD spatio-temporal variability at the fire patch level (i.e. assign individual fires from RSD to GBD) in order to more precisely quantify the dataset accuracy at the fire scale.

L279-280: Overall, RSD contain only a small fraction of the total number of fires documented by GBD.

L280-281: However, they capture reasonably well the temporal variability of fire activity across monthly and annual scales.

L294-296: Our findings suggested that global RSD of individual fires can be used to proxy variations in fire activity on monthly or annual timescales, however caution is advised when drawing from smaller fires (< 100 ha) across the Mediterranean region.

L296-297: Fire agencies may also benefit from the spatial and temporal consistency of remote-sensing data to support their operational fire mapping system at regional/national level.

[revised manuscript text omitted]

AltThough very promising, remote-sensing RS-datasets of individual fires have been sparingly compared to historical ground-based fire databases, that are generally thought to be- the most reliable source of data regarding fire occurrence and fire extent (Moreira et al., 2011; Mouillot et al., 2014). Previous studies indicated that rigorous evaluation of satellite estimates data with ground-based data is needed to assess the reliability of the RS information at regional scale (Turco et al., 2019). Most validation procedures of these remote-sensing RS- datasets were based on comparisons between different satellite products (Andela et al., 2019b; Laurent et al., 2018a), with however scarce attention to independent ground-based observations (Artés et al., 2019). In this work, we compared for the first time the three most recent remote-sensing RS-datasets of individual fires (Fire Atlas, FRY and GlobFire) with high qualityquality-controlled fire databases compiled by regional agencies across the most active fire region in Europe (i.e. Southwestern Mediterranean basin) during the common period of observations (2005 to 2015). While most previous studies have evaluated remote-sensing data on a fire-by-fire basis, this study aggregates individual fires across months and pixels (0.25°) and seeks to estimate how muchto what extent the temporal variability in both fire frequency and burned area are captured by remote-sensing datasets. We sought to provide a solid answer to the following questions. (i) Are remote-sensing RS-datasets capturing the actual pattern of fire occurrence and burned area? (ii) To what extent is their accuracy dependent on fire size? (iii) Can we rely on remote-sensing RS datasets to analyze fire regimes? To answer these questions, we assessed boththe spatial and temporal uncertaintyies in respect to BA and NF of estimated remote-sensing examined the agreement between remote-sensing and ground-based datasetsRS BA and NF aggregated at monthly and 0.25° resolutions across a range of individual fire size thresholds (1 to 500 ha). This study may inform end-users about remote-sensing datasetsRS ability to proxy actual fire activity but also on their limitations., and provide guidance on the correct usage of global fire datasetsRS information at regional scale.

**2 Data and Methods**

**2.1 Ground-based fire data**

The ground-based dataset (GBD) was built from multiple fire agencies sourcesThe ground based dataset was built from multiple regional/national sources, including fire records from Portugal, Spain, France and Sardinia in Italy (Table 1). All these ground monitoring systems provide high-quality datasets that have been extensively used in previous studies across France (Curt et al., 2014), Portugal (Pereira et al., 2011), Sardinia (Salis et al., 2013) and the Mediterranean basin (Rodrigues et al., 2020; Turco et al., 2016). Although not free of errors, these datasets constitute the most accurate source of historical information about fires available acrossin Europethe region.

Table 1. Description of Fregional fire agencies and reference link to the data used to build the harmonized ground-based database dataset (GBD) across the Southwest Mediterranean basin.

| Agency | Country | Coverage | Reference link |
| --- | --- | --- | --- |

| | | | |
|---|---|---|---|
| DECIF | Portugal | National | http://www2.icnf.pt/portal/florestas/dfci/relat/rel-if

 (last access: 10 January 2020) |
| EGIF | Spain | National | https://www.mapa.gob.es/va/desarrollo-rural/estadisticas/Incendios_default.aspx
 (last access: 18 December 2019) |
| Prométhée | France | Regional | https://www.promethee.com/
 (last access: 16 December 2019) |
| Regione Sardegna | Italy | Regional | http://webgis2.regione.sardegna.it/download/
 (last access: 22 January 2020) |

215  We extracted the following information from each regional datasets: the day of ignition, the fire size, and location of each fire. To ensure consistency across regions and scales, we analyzed the overlapping recording period among the datasets, i.e., 2005–2015. Small fires (<1 ha) were discarded to ensure the coherence of the analysis since these were not reported systematically by agencies over the studied period. The harmonized database contained 95,561 fire records, including only events that required
220 a firefighting response (i.e., disregarding agricultural and prescribed fires) (see Fig. 1).

[Figure]

**Figure 2.** (a) Mean annual burned area (BA, depicted by circle size) and mean annual number of fires (NF, depicted by color) at 0.25°
resolution over the study period (2005-2015).  (b) Spatial extent of the study area.

**2.2 Remotely-sensed fire data**

We used the most recent remote-sensing datasets (RSD)  of individual fires: Fire Atlas (Andela et al., 2019a, 2019b),
FRY (Laurent et al., 2018a, 2018b) and GlobFire (Artés et al., 2019; Artés Vivancos and San-Miguel-Ayanz, 2018). These
datasets provide the date and the spatial extent of individual fires from the pixel-based burned area MODIS product MCD64A1
Collection 6 (Table 2). The Terra and Aqua combined MCD64A1 is derived from the surface reflectance imagery and active

fires observation. It provides a global coverage of burned area estimation at a resolution of 500 m (Giglio et al., 2018). Fires were individualized from different algorithms such as a progression-based algorithm (Andela et al., 2019), a flood-fill algorithm (Laurent et al., 2018), and data mining (Artés et al., 2019) that share a common objective: assemble burned pixels that were adjacent in both space and time to identify and outline individual

235   fire events. All RSD  provide fire starting and ending dates, location and the final burned area for each retrieved fire event.

A key parameter of these algorithms is the cut-off value, which is defined as the maximum burn date difference allowed between two neighbouring pixels to be considered as belonging to the same fire event. This cut-off influences the size, shape and the degree of clumpiness and fragmentation of individual fire events (Laurent et al., 2018a; Oom et al., 2016). Fire Atlas

240   used spatially varying cut-off thresholds (4 to 10 days) depending on the fire frequency (Andela et al., 2019b), while the FRY algorithm processed four different cut-off scenarios (3, 5, 9 and 14 days), used in previous studies (Archibald and Roy, 2009; Hantson et al., 2015; Nogueira et al., 2017). Finally, GlobFire defined a fire event as a set of burned pixels that are connected within a 5-day window and have not been burned over the 16 previous days (Artés et al., 2019). For simplicity, we only reported the FRY cut-off value that performed the best (5 days). The comparison with all FRY cut-off values is available in

245   the Appendix A  (Fig A1).

**Table 2.** Description of the datasets  (RSD)  of individual fires, including the digital object identifier (DOI) and reference of each dataset. FA: Fire Atlas; FRY_M05: FRY MODIS (5 days) and GF: GlobFire.

| RSD  | Methodology | Cut-off values | Period | Dataset DOI | Reference |
|---|---|---|---|---|---|
| FA | Progression-based algorithm | 4 to 10 days | 2003-2016 | https://doi.org/10.3334/ORNLDAAC/1642 | (Andela et al., 2019b, 2019a) |
| FRY_M05 | Flood-fill algorithm | 5 days | 2000-2017 | https://doi.org/10.15148/0e999ffc-e220-41ac-ac85-76e92ecd0320 | (Laurent et al., 2018a, 2018b) |
| GF | Data mining | 5 and 16 days | 2000-2019 | https://doi.org/10.1594/PANGAEA.895835 | (Artés et al., 2019; Artés Vivancos and San-Miguel-Ayanz, 2018) |

250   **2.3   Methodology**

We compared burned area (BA) and number of fires (NF) estimated by RSD , with the ground-based reference GBD  ( Fig. 2). Only the common period between RSD  and GBD records (2005–2015) has been considered . We evaluated the ability of RSD  to reproduce the  temporal and spatial patterns of fire activity observed in GBD by fitting ordinary least squares (OLS) linear

255   regressions and using different metrics (OLS slope, R-squared correlation, and bias) to measure RSD accuracy. We calculated

the relative error (ε) as:

$$\varepsilon = 100 \times \frac{BA_{RSD} - BA_{AGD}}{BA_{AGD}} \qquad\qquad\qquad\qquad\qquad\qquad\qquad (1)$$

260

where, $BA_{RSD}$ represents the BA detected by remote-sensing datasets (RSD) and $BA_{AGD}$ represents the BA registered in the fire agencies datasets (GBD) over the study period. The analysis was repeated for the number of fires (NF).

265

We applied a land cover filter to the RSD data using CORINE Land Cover (CLC)  to exclude fires located within agricultural or artificial lands that are not always reported by fire agencies. To account for the land cover changes over the study period, we used CLC 2006 as a reference to filter RSD from the 2005-2009 period and CLC 2012 from 2010-2015. Sensitivity analysis to the land-cover filter is shown in the Appendix A (Fig. A2).

270 As RSD  are prone to omit smaller fires (<25 ha) due to the coarse spatial resolution of MODIS product MCD64A1 (500 m) and other limitations, we investigated different fire size thresholds increasing from 1 to 500 ha. Analyses were repeated for each size-filtered sample (i.e. excluding fires smaller than a given threshold).

[Figure]

**Figure 2.** The general framework for comparison of RS  with GBD in terms of burned area (BA) and number of fires (NF)  across a range of individual fire size thresholds (1 to 500 ha).

**2.3.1 Temporal agreement**

280  All datasets were aggregated to monthly scale over the whole study area. We retrieved the slope coefficient of OLS regressions and the coefficient of determination (R-squared) as a proxy of agreement between RS and GBD. Slope values greater than 1 indicated an underestimation of fire activity as seen by GBD  and vice versa. A slope equal to 1 would imply

a perfect agreement.

285

$$\varepsilon = 100 \times \frac{RS - AG}{AG} \tag{1}$$

**2.3.2 Spatial agreement**

We then sought to examine how the agreement between RSD and GBD datasets varies across space. There is much uncertainty in estimating the ignition point from satellite data, mainly due to the spatial and temporal proximity of fire pixels and the possibility of multiple ignition points in a single fire event (Benali et al., 2016). Likewise, GBD  do not provide systematically ignition points. Thus, to overcome this limitation, we aggregated both RS and GBD  onto a 0.25° grid (≈ 25 km), setting a common ground for both datasets.

To examine the spatial agreement between RS and GBD, we calculated the relative error (Eq. 1) for each grid cell. Finally, we estimated the overall spatial error, computed as the ε averaged across all grid cells for each dataset.

**3  Results**

**3.1  Temporal agreement**

We first analyzed the monthly distributions of BA and NF  for all fires (>1 ha) aggregated across the whole studied area. Fig. 3 shows that RS  follow a similar variability in terms of monthly BA but systematically underestimate BA and NF with respect to GBD. The best agreement between RS  and GBD occurs mainly during the warm season (May to October; see Fig. 4). This is usually the period experiencing the largest fires, which account for the bulk of BA in the region (Turco et al., 2016). Conversely, the poorest agreement was found during the cool season (November to April), a period dominated mainly by small fires linked to agricultural activities.

[Figure]

**Figure 3.** (a) Monthly burned area and (b) number of fires (>1 ha) in each fire dataset across the Southwest Mediterranean basin over 2005-2015.

[Figure]

310

**Figure 4.** (a)  Seasonal monthly error (ε) for burned area and (b) number of fires estimates of each RSD  for all fires >1 ha in the studied area. Cool season from November to April and warm season from May to October. Dashed lines represent the perfect agreement between the datasets.

Table 3 presents the total BA and NF as well as monthly (i.e. including the seasonal cycle) and annual correlation (i.e. excluding the seasonal cycle) between RSD and GBD for all fires (>1 ha). Monthly correlations showed a stronger agreement for BA ($R^2 \approx 0.98$) than for NF ($R^2 \approx 0.89$). Annual correlations, where the effect of the seasonal cycle was removed, also showed very high values ($R^2 \approx 0.99$). Despite the fact that RSD underestimated the total BA by 38% and the NF by 96% for all fires, they reproduced almost perfectly the temporal variability on both monthly and annual basis. The difference in absolute numbers thus relates to undetected small fires in RSD .

**Table 3.** C correlation between RSD and GBD of monthly and annual burned area and number of fires  for all fires (>1 ha) between 2005 and 2015 .

| Dataset | Burned area | | | Number of fires | | |
|---|---|---|---|---|---|---|
| | Total (ha) | Mo. correlation | Yr. correlation | Total (n) | Mo. correlation | Yr. correlation |
| AGENCIES | 2,527,603 | - | - | 95,561 | - | - |
| FA | 1,609,267 | 0.99 | 0.99 | 3,875 | 0.90 | 0.99 |
| FRY_M05 | 1,524,171 | 0.99 | 0.99 | 2,134 | 0.88 | 0.99 |
| GF | 1,562,001 | 0.98 | 0.99 | 4,637 | 0.90 | 0.99 |

325 The monthly agreement of BA and NF (Fig. 5) strongly varies with fire size thresholds (1, 50, 100 and 500 ha). The positive slope of the linear trends indicates that RSD generally underestimate both BA and NF when accounting for all fires (> 1 ha). However, they become progressively more accurate as the fire size threshold increases, a feature that is particularly evident in NF estimates (Fig. 5 e-h).

[Figure]

**Figure 5.** Comparison of GBD and RS in respect to monthly burned area (top) and the number of fires (bottom) when considering a) all fires (> 1 ha), b) fires >50 ha, fires >100 ha and d) fires >500 ha. (e-h) Same as a-d) but for the number of fires. The 1:1 dashed lines represent the perfect fit between the datasets.

Fig. 6 shows the evaluation of RS  through different metrics over the continuum of fire size thresholds. Except for the R-Squared (Fig. 6, middle) which saturates for fires >100 ha for NF, all metrics present a similar behavior showing better

agreement when increasing the fire size threshold. Overall, BA (Fig. 6, top) presented better accuracy than NF (Fig. 6, bottom). Despite the different methodologies used to reconstruct individual fires, all datasets showed similar scores, albeit FA displayed lower relative error ($\varepsilon$) for NF.

[Figure]

[Figure]

[Figure]

**Figure 6.** Evaluation of RSD̶ ̶d̶a̶t̶a̶s̶e̶t̶s̶ through different metrics including the slope (left), R-squared correlation (middle) and relative error (right) for both burned area (top) and the number of fires (bottom) over a range of individual fire size thresholds (1 to 500 ha). Dashed lines indicate a perfect fit between RS and AG fire data.

**3.2 Spatial agreement**

345   Fig. 7 shows the spatial distribution of the relative error ($\varepsilon$) for BA over different individual fire size thresholds (for all fire size thresholds see Supplementary material̶D̶a̶t̶a̶). As expected from previous results, RSD̶ ̶d̶a̶t̶a̶s̶e̶t̶s̶ strongly underestimated BA, especially when including smaller fires. However, a few exceptions are seen for fires < 50 ha mainly over eastern Spain, suggesting that RSD detect in that case more fires than A̶G̶GBD. This may be related to a few and small prescribed fires that we̶a̶re not reported in A̶G̶GBD. Also, we found much lower $\varepsilon$ ̶r̶e̶l̶a̶t̶i̶v̶e̶ ̶e̶r̶r̶o̶r̶s̶ in regions with higher fire activity, such as the
350   Northern Iberian Peninsula. This is rather expected, as an absolute change in regions with high (low) baseline will result into a small (large) percentage change.

[Figure]

[Figure]

**Figure 7.** The relative error (ε) of the total burned area computed as the relative difference between RSD and BD data over different individual fire size thresholds (1, 50, 100 and 500 ha). The overall ε is indicated on each map.

Likewise, RSD strongly underestimated NF (Fig. 8), likely disregarding those smaller fires not detected by MODIS. Surprisingly, a few areas showed positive differences in NF for fires >100 ha across parts of Spain. This overestimation of large fires may be related to the fact that RSD algorithms are likely to split larger fires into multiple events. Nevertheless, the overall relative error between RSD and BD decreases when focussing on larger fires for both NF and BA, highlighting the important role of fire size on RSD accuracy.

[Figure]

[Figure]

**Figure 8.** Same as Fig. 7 but for number of fires.

**4    Discussion**

nderstand_ing_ global changes in fire activity calls for efficient and harmonized approaches to record fire activity. Satellite-borne spectral and thermal sensors offer several global fire products, evolving from BA mapping and active fire detection to novel developments post-processing BA products into single fire datasets (Chuvieco et al., 2019). The ongoing challenge lies in determining their reliability and usefulness. Here, we compared RS_D_ with GBD across the Southwestern Mediterranean basin to better understand RS_D_  limitations and guide end-users.

Although RSD may miss a substantial number of fires, the temporal variations in both NF and BA match very well with ground-based observations. Our results also demonstrate that agreement between RSD and GBD is strongly dependent on individual fire size. Focusing on larger fires (fire typically

375   > 100 ha), RS were in stronger agreement with GBD regardless of the evaluated metrics. Fires > 100 ha denoted much lower error (BA 10%; NF 35%), especially in regions with higher fire activity such as the northwest of the Iberian Peninsula or the south of Sardinia . Our findings are in agreement with previous studies, which pointed at fire size as the primary limiting factor for remotely-sensed fire data  (Campagnolo et al., 2021; Rodrigues et al., 2019; Ying et al., 2019; Zhu et

380   al., 2017).

    The ability of RS to identify individual fires depends mainly on two features: the processing algorithm and the underlying reliability of the BA product. The relatively low capacity of the latter to detect small fires is related to the coarse spatial resolution (500 m) of the MODIS sensor. Several recent studies have shown that MODIS products rather reliably detect fires over 40–120 ha but miss a number of smaller fires (Fusco et al., 2019; Giglio et al., 2018; Rodrigues et al., 2019; Zhu et

385   al., 2017). Although other BA products, such as FireCCI50 (Chuvieco et al., 2018), provide finer spatial resolution (250 m), a substantial number of small and/or highly fragmented fires remain undetected, leading to a considerable underestimation of BA (Roteta et al., 2019). In addition, all space-borne BA products face many other well-documented limitations such as the variability in orbital coverage, satellite overpass time, and satellite view obstruction (Cardoso et al., 2005; Padilla et al., 2014). In this sense, detectability may vary regionally across the globe and without ground-based fire datasets, it may be difficult to

390   properly validate their reliability (Turco et al., 2019). Nonetheless, the limitations of MCD64A1 are inherent to all RS, since all of the analyzed products were built on this basis. Hence, differences among RS are rather expected to be associated with the underlying algorithm used to identify single fire events.

    RS were found to better  estimate BA than NF. This disparity relies on the complexity of extracting individual fires from gridded BA products. Environmental conditions (e.g. topography, cloud/smoke cover) may influence the

395   sensor detection power, resulting in a break in BA continuity thereby increasing the risk of artificially splitting single fires into different fire events.. Likewise, if a fire lasts longer than the defined cut-off window, it will be automatically split into different events (Oom et al., 2016). In addition, if multiple fires occur simultaneously in the same region, the parameterization of the RS algorithms may merge multiple individual fires

400   (Archibald et al., 2013). Lastly, regional features of the fire regime may constrain RS accuracy. For instance, the Mediterranean fire regime is known for hosting numerous small fires, which are unlikely to be detected by satellite observations (Turco et al., 2016). These fires do not contribute very much to the total annual burned area but significantly harm the performance of the RS in terms of NF .

    The selection of an appropriate fire size threshold depends on the objectives of each analysis. However, in this study~~Even

405   though the selection of an appropriate fire size threshold depends on the objectives of each analysisD datasetss~~ from the use of a spatially explicit cut-off threshold,

taking both fire spread rate and satellite coverage into account to track the extent of individual fires (Andela et al., 2019b).

410 However, uncertainty in MODIS largely outpaces the uncertainties across the RSD datasets. The low capacity of gridded BA products to detect small-mid fire events (< 100 ha) can be improved by the generation of products based on higher resolution sensors in the range of 10–30m (Roteta et al., 2019). RSD of individual fires derived from finer gridded BA would provide better accuracy in the fire metrics, specifically for NF. In addition, the MCD64A1 product already incorporates the uncertainty of detection as an auxiliary variable of gridded BA data (Giglio et al., 2018). RSD could benefit from this and report similar

415 information at individual fire level.

The spatio-temporal aggregation applied in our study is expected to increase the signal-to-noise ratio and thus decrease the uncertainty in RSD estimates. According to Turco (2019), the spatial agreement between remotely-sensed and ground based fire data AG and RS increases at lower resolutions, being generally best when aggregating the data onto a 1° grid (approximately 110 km) or beyond. Likewise, aggregating the data over time (either monthly or annually) also increases the

420 signal-to-noise ratio by filtering out the temporal stochastic noise (Spadavecchia and Williams, 2009). Evaluating RSD datasets on shorter timescales and/or finer spatial resolutions would likely deteriorate the agreement with AGGBD. Nevertheless, thea spatio-temporal aggregation, such as the one employed here, has been extensively used in previous studies analyzing fire regimes at regional (Barbero et al., 2014; Jiménez-Ruano et al., 2020; Parisien et al., 2014) and global scales (Bedia et al., 2015; Di Giuseppe et al., 2016; Turco et al., 2018b).

425 Further studies are still needed to examine RSD spatio-temporal variabilityestimates at the fire patch level (i.e. assign individual fires from RSD to AGGBD) in order to more precisely quantify RS the dataset accuracy at the fire scale.

**5 Data availability**

The above described fire datasets, their characteristics and reference to access the data can be found in Tables 1 and 2. All these fire datasets are open access except one of the ground-based datasets (EGIF) that is available upon request. The different

430 data producers host the data in different ways, typically using websites or data repositories. The harmonized AGGBD database used here as ground-based reference is available at https://doi.org/10.5281/zenodo.3905040 (Galizia et al., 2020).

**6 Conclusion**

In this work, we built upon previous research and investigated the reliability of three RSD datasets of individual fires over a range of fire size thresholds across the Southwestern Mediterranean basin. Overall, RSD contain only a small fraction of the

435 total number of fires documented by GBD. However, they RS datasets were able to capture reasonably well the temporal variability of and spatial patterns of fire activity across monthly and annual scales. , with however limited ability to outline small to mid fire events. Despite the different methodologies used to reconstruct fire patches, all datasets RSD (FA, FRY and

[revised manuscript text omitted]